# Multi-Scale Spatial-Temporal Hypergraph Network with Lead-Lag Structures for Stock Time Series Forecasting

## Abstract

Time series forecasting occurs in a range of financial applications providing essential decision-making support to investors, regulatory institutions, and analysts. Unlike multivariate time series from other domains, stock time series exhibit industry correlation. Exploiting this kind of correlation can improve forecasting accuracy. However, existing methods based on hypergraphs can only capture industry correlation relatively superficially. These methods face two key limitations: they do not fully consider inter-industry lead-lag interactions, and they do not model multi-scale information within and among industries. This study proposes the **Hermes** framework for stock time series forecasting that aims to improve the exploitation of industry correlation by eliminating these limitations. The framework integrates moving aggregation and multi-scale fusion modules in a hypergraph network. Specifically, to more flexibly capture the lead-lag relationships among industries, Hermes proposes a hyperedge-based moving aggregation module. This module incorporates a sliding window and utilizes dynamic temporal aggregation operations to consider lead-lag dependencies among industries. Additionally, to effectively model multi-scale information, Hermes employs cross-scale, edge-to-edge message passing to integrate information from different scales while maintaining the consistency of each scale. Experimental results on multiple real-world stock datasets show that Hermes outperforms existing state-of-the-art methods in both efficiency and accuracy. All datasets and code are available at https://anonymous.4open.science/r/Hermes-E150.

## 1 Introduction

Stock time series are typically composed of data from multiple stocks, each of which includes multiple key indicators such as stock opening prices, closing prices, etc. (Fan & Shen, 2024; Li et al., 2024). Stock time series forecasting (STSF) is important in financial applications, providing crucial decision-making support for investors, regulatory institutions, and analysts, helping them make data-driven decisions in complex and ever-changing market environments (Rather et al., 2017; Faloutsos et al., 2018; Jacob & Shasha, 1999; Zhang et al., 2014; Zheng et al., 2023).

Most existing studies treat stocks in stock time series independently, thus disregarding the interconnected nature of market dynamics and contradicting the intrinsic interdependencies observed in real-world financial systems (Diebold & Yılmaz, 2014). In reality, stocks are often interrelated, and there are rich signals in the relationships between them (Nobi et al., 2014). For example, with the breakthroughs of Artificial Intelligence technology, companies like Apple, Microsoft, and Alphabet that actively invest and innovate in this domain are expected to lead in the market due to their technological advantages. It is one of the important factors for their stock prices to show a steady upward trend in the overall stock market[1]—see Figure 1a. At the same time, the soaring power demand, driven by electrification, decarbonization, and the advent of artificial intelligence, led to a concurrent rise in the stock prices of companies in the energy sector[2]—see Figure 1a. Such industry correlations have gradually attracted more attention. Especially within the same industry, different

---

[1] www.morningstar.co.uk/uk/news/258865/tech-and-communication-stocks-drove-us-market-gains-in-2024.aspx
[2] https://finance.yahoo.com/news/american-electric-power-aep-among-121752028.html?

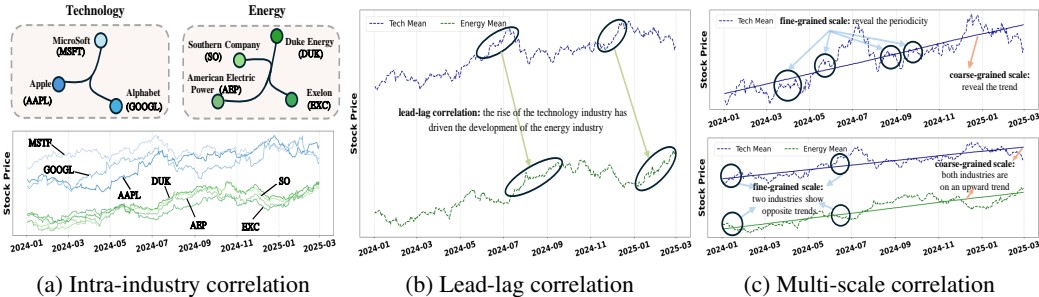

(a) Intra-industry correlation        (b) Lead-lag correlation        (c) Multi-scale correlation

Figure 1: (a): Intra-industry correlation. Relations between stocks within an industry (hyperedge) can be represented by a hypergraph. (b): Lead-lag correlation. (c): Multi-scale correlation.

stocks often exhibit correlations, reflecting the common economic environment, market fluctuations, policy changes, and other factors that impact the stocks.

Hypergraphs can connect multiple graph nodes (representing, e.g., stocks) through hyperedges. By connecting stocks within the same industry by hyperedges to construct a hypergraph network, it is possible to capture more comprehensively the interrelationships and dynamics among stocks within an industry. However, existing hypergraph-based STSF methods that consider industry correlations still have the following challenges.

**Challenge 1: Current methods fail to fully consider inter-industry lead-lag interactions.** In financial markets, there are causal relationships and time dependencies among industries, and these relationships are highly diverse and complex. As shown in Figure 1b, the technology industry has flourished with breakthroughs in key technologies and market innovations, achieving significant progress. Technological innovations from technology companies have injected new vitality into the development of renewable energy, and energy companies have leveraged technologies such as big data, AI, and the Internet of Things to enhance the efficiency of energy production and management[2], thereby driving up stock prices (Abid et al., 2025; Olanrewaju et al., 2024; Rafiei et al., 2025). Therefore, in the modeling process, it is important to capture lead-lag correlations among industries more accurately, so as to better reflect the dynamic interactions and dependencies in financial markets, ultimately improving the accuracy and practicality of the forecasts.

**Challenge 2: Current methods do not model multi-scale information within and among industries.** Stock time series inherently possess multi-scale characteristics, and markets exhibit different features and patterns at different time scales. For example, in Figure 1c, we can observe a clear overall trend at a coarse-grained scale, and a cyclical pattern at a fine-grained scale. Moreover, two different industries, may exhibit correlations at a coarse-grained scale. However, delving deeper into a fine-grained analysis might find that two industries show opposite trends. Single-scale analysis methods cannot fully capture such characteristics, so there is a pressing need to adopt multi-scale analysis to reveal the dynamic changes in financial markets across different time scales, thereby providing richer and more accurate information for stock time series forecasting.

In this study, we address the two challenges by proposing a general stock time series forecasting framework, **Hermes**, which integrates *hyperedge-based moving aggregation* and *hyperedge-based multi-scale fusion* modules into a spatial-temporal hypergraph network. These integrations enhance the precision and stability of stock forecasting. First, to more flexibly model lead-lag relationships among industries, we integrate a *hyperedge-based moving aggregation module* into the hypergraph network. This module incorporates a sliding window and utilizes dynamic temporal aggregation operations to consider lead-lag dependencies among industries. This approach aims to capture a market's complex structure more precisely. Second, to address the issue that existing methods fail to effectively model multi-scale information, we decompose raw data at multiple scales and integrate a *multi-scale fusion module* into the hypergraph network. This module employs cross-scale, edge-to-edge message passing to integrate information from different scales while preserving the consistency of each scale, thereby boosting both the accuracy and stability of forecasts. Experimental results on multiple real-world stock datasets show that Hermes outperforms existing state-of-the-art methods in both efficiency and accuracy. Our contributions are summarized as follows.

- To solve the STSF problem, we propose a general spatial-temporal hypergraph network framework, **Hermes**, which learns more accurate and adaptive forecasting models by considering both lead-lag and multi-scale industry correlations.

- We design a hyperedge-based multi-scale fusion module. This module utilizes a cross-scale, egde-to-edge message-passing to integrate information from different scales while maintaining the consistency of each scale.

- We propose a hyperedge-based moving aggregation module that introduces a sliding window and utilizes dynamic temporal aggregation operations to consider lead-lag dependencies among industries.

- We report on extensive experiments on public stock datasets, finding that Hermes is capable of outperforming state-of-the-art baselines.

## 2    RELATED WORK

### 2.1    DEEP LEARNING STOCK TIME SERIES FORECASTING

With the rise of deep learning, notable progress has occurred in stock forecasting. This is mainly because deep neural networks can model complex nonlinear patterns. Some studies use recurrent neural networks (RNNs) (Nelson et al., 2017; Qin et al., 2017; Akita et al., 2016; Rahman et al., 2019) to capture the historical development in individual stock prices and predict their short-term trends. In addition, due to the significant progress of Multi-Layer Perceptrons (MLPs) architecture in the general time series field (Zeng et al., 2023; Lin et al., 2025; 2024). Increasingly may proposals target stock forecasting performance by improving the MLPs architecture. For example, a series of studies (Fan & Shen, 2024; Lazcano et al., 2024; Tashakkori et al., 2024) enhance the learning ability of simple MLPs structures by utilizing the MLP-Mixer backbone, thereby improving forecasting accuracy. Next, the Transformer model has become increasingly popular due to the ability of its self-attention mechanism to capture long-term dependencies. For example, Master (Li et al., 2024) proposes a novel stock transformer for stock price forecasting to effectively capture stock correlation. While these deep learning methods have advanced stock forecasting, the complexity of stock data and the need to account for intricate relationships among stocks have led researchers to explore new methods based on hypergraphs.

### 2.2    HYPERGRAPH-BASED STOCK TIME SERIES FORECASTING

Traditional graphs have limitations when it comes to modeling higher-order multivariate relationships, as they can only represent relations between pairs of nodes. To address this issue, Hypergraph Neural Networks (HGNNs) (Feng et al., 2019c) were introduced. Hypergraphs extend the concept of simple graphs to enable the capture of relationships among multiple stocks (Chen et al., 2020). Hypergraphs have gained increasing attention and application across various fields. Hyperedge in a hypergraph represents a set of vertices, making them suitable for modeling non-pairwise relationships (Luo et al., 2014; Sawhney et al., 2020; Huynh et al., 2023). For example, HGAM (Li et al., 2022) is a hypergraph-based reinforcement learning method for stock portfolio selection. THGNN (Xiang et al., 2022) is a temporal and heterogeneous graph neural network model that aims to learn dynamic relationships by using two-stage attention mechanisms. STHAN-SR (Sawhney et al., 2021) enhances corporate relevance based on Wiki data and uses hypergraph convolution to propagate information from higher-order neighbors. ESTIMATE (Huynh et al., 2023), employs hypergraphs to capture non-pairwise correlations, utilizing temporal generative filters to identify individual patterns for each stock. Unlike the above methods, we propose a new hypergraph-based stock time series forecasting model that specifically takes into account the multi-scale nature of stock data, enabling it to capture the complexity of market dynamics more comprehensively. Furthermore, our approach places particular emphasis on the lead-lag relationships between financial industries, where the changes in one industry may precede or lag behind those in another. By capturing these key relationships more fully, we aim to improve forecasting accuracy.

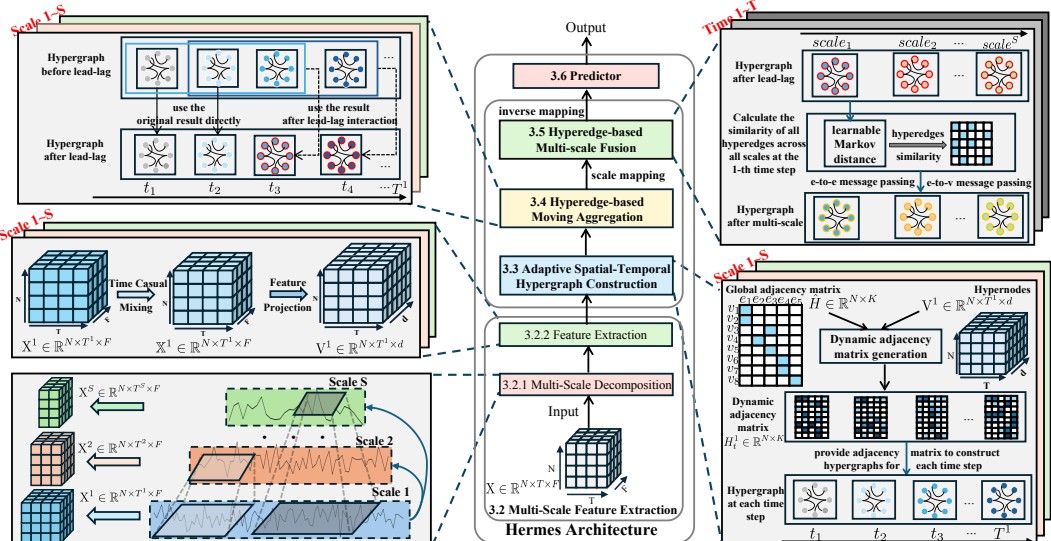

Figure 2: The architecture of Hermes.

## 3 METHODOLOGY

In financial markets, a stock time series $\mathbf{X} = \{\boldsymbol{X}_1, \ldots, \boldsymbol{X}_N\}$ records the historical data of $N$ stocks, where $\boldsymbol{X}_i \in \mathbb{R}^{T \times F}$ is data of one stock with a lookback window of $T$ time steps and with $F$ technical indicators at each time step. In stock time series, the indicators may include opening price, closing price, highest price, lowest price, and trading volume. Following existing studies (Huynh et al., 2023; Fan & Shen, 2024), we input a stock time series with multiple indicators, denoted as $\mathbf{X} \in \mathbb{R}^{N \times T \times F}$, and aim to predict the closing price $p_i^t$ at time step $t$. We define the 1-day return ratio as $r_i^t = \frac{p_i^t - p_{i-1}^t}{p_{i-1}^t}$. With $\theta$ representing the model parameters, the process can be expressed as follows.

$$\mathbf{X} \in \mathbb{R}^{N \times T \times F} \xrightarrow{\theta} p \in \mathbb{R}^{N \times 1} \to r \in \mathbb{R}^{N \times 1} \tag{1}$$

### 3.1 FRAMEWORK OVERVIEW

In this section, we introduce the framework of our proposed spatiotemporal hypergraph neural network that considers multiple time scales and models the lead-lag relationships between industries. Our framework takes historical financial sequences as input, first samples the time series into several sequences of different granularities according to different time scales, and maps the feature dimension to the hidden dimension to consider the high-dimensional correlations between time series segments under different scales. Subsequently, we consider constructing relationships between agents at each scale, specifically considering constructing hyperedges for agents in the same industry, and modeling the lead-lag relationships between industries in the corresponding hypergraphs. Then, we model the interactions between different scales, construct hyperedges for industries of different scales, and adaptively learn the correlations between them under the corresponding multi-scale hypergraph, thereby playing a role in fusing scales. Finally, we predict the corresponding indicators for each agent for the next day through a prediction head based on a multilayer perceptron.

### 3.2 MULTI-SCALE FEATURE EXTRACTION

**Multi-Scale Decomposition.** Time series of different scales exhibit different characteristics, among which fine scales primarily describe detailed patterns and contain more periodic information, while coarse scales highlight macro changes and cover the overall trend changes. This multi-scale perspective can essentially unravel the complex variations within multiple components, thereby facilitating time variation modeling. It is worth noting that, especially for prediction tasks, due to their different dominant time patterns, multi-scale time series show different predictive capabilities.

Specifically, for the financial input time series $\mathbf{X} \in \mathbb{R}^{N \times T \times F}$, which denotes the records of $F$ key indicators for $N$ market agents of $T$ timestamps in the past. To decouple the intricate multi-resolution information, we use convolutional layers of different sizes to adaptively down-sample the time series into sequences of different scales:

$$\mathbf{X}^i = \text{1D-Conv}^i(\mathbf{X}), \tag{2}$$

where $\mathbf{X}^i \in \mathbb{R}^{N \times T^i \times F}$ denotes the $i$-th scale of financial time series, $T^i$ is the down-sampled length. We then obtain the multi-scale time series sequences $\{\mathbf{X}^1, \mathbf{X}^2, \cdots, \mathbf{X}^S\}$ of $S$ kinds of scales through $S$ convolutional layers with different kernel sizes and stride lengths.

**Feature Extraction.** Subsequently, for time series of different scales, we enhance their representations at two aspects. First, we adopt the Causal-Mixing technique to process the temporal dimensional. The Causal-Mixing technique ensures causal relationships rather than full connectivity, and ensure that time series points can only see previous points. It consists of a group of MLP layers:

$$\text{Causal-Mixing}^i = \{\text{MLP}^{i,1}, \text{MLP}^{i,2}, \cdots, \text{MLP}^{i,T^i}\}, \ \ \text{MLP}^{i,j} = \mathbb{R}^j \to \mathbb{R} \tag{3}$$

$$\mathbb{X}^i = \text{Causal-Mixing}^i(\mathbf{X}^i), \tag{4}$$

the Causal-Mixing is applied along the temporal dimensional, which maps $X^i_{1:j}$ through $\text{MLP}^{i,j}$ to $\mathbb{X}^i_j$. The new representation $\mathbb{X} \in \mathbb{R}^{N \times T^i \times F}$ keeps the original shape.

Then their feature dimensions are mapped into the high-dimension hidden spaces to enrich their representation capabilities, facilitating the extraction of more complex hidden relationships:

$$\mathcal{X}^i = \mathbb{X}^i \cdot W^i, \tag{5}$$

where $\mathcal{X}^i \in \mathbb{R}^{N \times T^i \times d}$, $d$ is the dimension of the latent space. $W^i \in \mathbb{R}^{F \times d}$, constructing a linear projection to enhance representational capability.

### 3.3 Adaptive Spatial-Temporal Hypergraph Construction

For the representation $\mathcal{X}^i \in \mathbb{R}^{N \times T^i \times d}$ of $i$-th scale, the hypergraph of $i$-th scale $\mathcal{X}^i$ can be constructed through a global-shared matrix $H \in \mathbb{R}^{N \times K}$, where $N$ represents the number of stocks, $K$ represents the number of industries, and $H_{l,m} = 1$ denotes that the $l$-th stock belongs to the $m$-th industry. The $i$-th Spatial-Temporal hypergraph $\mathcal{G}^i$ is formulated as:

$$\mathcal{G}^i = (H, \mathcal{X}^i, E^i), \tag{6}$$

where $\mathcal{X}^i$ is the representation of $i$-the scales and works as the nodes of the $\mathcal{G}^i$, $E^i \in \mathbb{R}^{K \times T^i \times d}$ is the hyperedge constructed through the stock-industry relationship matrix $H$:

$$\text{scores}^i = \text{Linear}^i(\mathcal{X}^i) \in \mathbb{R}^{N \times 1}, \ \ \text{Linear}^i = \mathbb{R}^{T^i \times d} \to \mathbb{R}, \tag{7}$$

$$H^i = H \odot \text{scores}^i + (1 - H) \odot (-\infty), \ \ \mathcal{H}^i = \text{SoftMax}(H^i, \dim = 0), \ \ E^i = (\mathcal{H}^i)^T \cdot \mathcal{X}^i, \tag{8}$$

where the matrix $H$ is first processed with a learnable renovation to adapt to the current representation $\mathcal{X}^i$. $H^i$ is the intermediate result obtained through endowing the $H_{l,m} = 1$ parts with learnable $\text{scores}^i$ and $H_{l,m} = 0$ parts with $-\infty$ to ensure zeros in the SoftMax step. Then the $\mathcal{H}^i \in \mathbb{R}^{N \times K}$ with Continuous Weights is obtained, and the hyperedges $E^i \in \mathbb{R}^{K \times T^i \times d}$ of industries are generated through information integration from all the related agents. We obtain all the Spatial-Temporal hypergraphs $\{\mathcal{G}^1, \mathcal{G}^2, \cdots \mathcal{G}^S\}$ for each scale as above mentioned.

### 3.4 Hyperedge-based Moving Aggregation

After delineating different time series scales with Spatial-Temporal hypergraphs $\{\mathcal{G}^1, \mathcal{G}^2, \cdots \mathcal{G}^S\}$, we further consider the lead-lag relationships between industries under each scale, to analyze the complex correlations in financial markets—see Figure 3 in Appendix B.4.

We construct a message passing mechanism on the hyperedges to capture the lead-lag relationships between industries. First, we employ a fixed-size sliding window as the range for examining the

lead-lag relationships. For each hyperedge, through the sliding window, we can delineate different patches, which serve as the smallest units for constructing the lead-lag relationships:

$$\mathcal{P}^i = \text{SlidingWindow}(E^i, \text{size} = k^i, \text{stride} = 1), \tag{9}$$

where $\mathcal{P}^i \in \mathbb{R}^{K \times (T^i - k^i + 1) \times k^i \times d}$ denotes the patches of $E^i$, containing local information with the size of $k^i$. From the perspective of causality, considering the relationship between the first few points and the last point within the window, we can construct the relationship matrix of the leading part and the lagging part. We then mine the lead-lag relationships inside each patch by considering the correlations between $\mathcal{P}^i_{before}$ and $\mathcal{P}^i_{last}$:

$$\mathcal{P}^i_{before} = \text{Reshape}(\mathcal{P})^i \in \mathbb{R}^{[K \times (T^i - k^i + 1)] \times k^i \times d}, \tag{10}$$

$$\mathcal{P}^i_{last} = \text{Reshape}(\mathcal{P})^i_{-1} \in \mathbb{R}^{(K \times 1) \times k^i \times d}, \tag{11}$$

Subsequently, by using the message passing method to propagate through the leading part, we obtain the feature fusion among hyperedges:

$$\tilde{A}^i = \mathcal{P}^i_{before} \otimes \mathcal{P}^i_{last} \in \mathbb{R}^{[K \times (T^i - k^i + 1)] \times K \times k^i}, \tag{12}$$

$$A^i = SoftMax(\tilde{A}^i, \text{dim} = 2), \;\; \tilde{E}^i = A^i \otimes \mathcal{P}^i_{before} \in \mathbb{R}^{K \times k^i \times d}, \tag{13}$$

$$\hat{E}^i = \text{MLP}^i(\tilde{E}^i) + E^i, \;\; \text{MLP}^i := \mathbb{R}^{k^i} \to \mathbb{R}^{T^i}, \;\; \mathcal{V}^{e^i} = \hat{E}^i \otimes \mathcal{H} \in \mathbb{R}^{N \times T^i \times d}, \tag{14}$$

where $\tilde{A}^i$ considers the correlations among the leading parts and lagging parts, $A^i$ probabilityizes $\tilde{A}^i$ to control weightsuming, $\hat{E}^i \in \mathbb{R}^{K \times T^i \times d}$ denotes the embedding of hyperedges integrated with lead-lag relationships among different industries. $\mathcal{V}^{e^i}$ is the updated nodes of $i$-th scale integrated with the lead-lag information extracted on hyperedges—see Figure 4 in Appendix B.4.

This is the method of considering the leading and lagging relationships between industries under each scale using Spatial-Temporal hypergraphs. After performing the above operation for each scale, we obtain a multi-scale representation of hyperedges $\{\hat{E}^1, \hat{E}^2, \cdots, \hat{E}^S\}$. Subsequently, fusion between scales will be carried out to integrate information of different granularities.

### 3.5 HYPEREDGE-BASED MULTI-SCALE FUSION

For multi-scale representation of hyperedges $\{\hat{E}^1, \hat{E}^2, \cdots, \hat{E}^S\}$ which contains the industry information, we further extract the potential correlations among different scales to reveal the latent interactions among different industries. First, we up-sample different scales back to the original length to ensure uniform representation size:

$$\hat{\mathcal{E}}^i = \text{Linear}^i(\hat{E}^i) \in \mathbb{R}^{K \times T \times d}, \;\; \text{Linear}^i = \mathbb{R}^{T^i} \to \mathbb{R}^T, \tag{15}$$

$$\hat{\mathcal{E}} = \{\hat{\mathcal{E}}^1, \hat{\mathcal{E}}^2, \cdots, \hat{\mathcal{E}}^S\} \in \mathbb{R}^{S \times K \times T \times d} \tag{16}$$

Then, we construct adaptive adjacency matrices for these hyperedges of different scales to measure the correlations between different industries at various information granularities. We employ metric learning for this task and use a learnable Mahalanobis distance to automatically build relationships between different industries in the hidden space:

$$\tilde{\mathcal{E}} = \text{Flatten}(\hat{\mathcal{E}}) \in \mathbb{R}^{(S \times K) \times T \times d}, \;\; D_{ij} = (\tilde{\mathcal{E}}_i - \tilde{\mathcal{E}}_j)^T \otimes Q \otimes (\tilde{\mathcal{E}}_i - \tilde{\mathcal{E}}_j), \tag{17}$$

$$C_{ij} = \begin{cases} \frac{1}{D_{ij}} & \text{if } i \neq j \\ 0 & \text{if } i = j \end{cases}, \;\; B = \text{SoftMax}(C, \text{dim} = 0) \tag{18}$$

where $D, C, B \in \mathbb{R}^{(S \times K) \times (S \times K) \times T \times d}$. $Q \in \mathbb{R}^{d \times d}$ is a learnable semi-positive definite matrix, and can be practically constructed by $Q = L^T \cdot L$. Subsequently, we construct interactions between hyperedges at different scales using message passing and further update the representation of hyperedges.

$$\tilde{B} = \mathcal{D}^{-1} \otimes B \otimes \mathcal{D}, \;\; \mathcal{E}' = \tilde{B} \otimes \tilde{\mathcal{E}} \otimes W^e, \tag{19}$$

where $\mathcal{D} \in \mathbb{R}^{(S \times K) \times T \times d}$ is the degree matrix, $W^e \in \mathbb{R}^{d \times d}$ is the learnable parameters in the graph message passing mechanism. After integrating the lead-lag relationships between industries

on the hyperedges and fusing the hidden relationships between different scales, the representation of the hyperedges $\mathcal{E}'$ is adaptively updated. We then continue to complete the message passing from hyperedges to nodes—see Figure 4 in Appendix B.4, thereby feeding back the complex correlations within the industry to specific agents and completing the update of the representation of agent nodes:

$$\mathcal{V} = \mathcal{E}' \otimes \mathcal{H} \in \mathbb{R}^{(S \times N) \times T \times d} \tag{20}$$

### 3.6 PREDICTOR

After completing the complex representation extraction using a hypergraph-based structure, we employ an MLP-based predictor and incorporate residual connection modules to forecast financial indicators across each scale $s$.

$$\hat{\mathcal{X}}^s = \text{MLP}_{s,1}(\mathcal{X}^s) \in \mathbb{R}^{N \times 1}, \ \ \tilde{\mathcal{V}}^s = \text{MLP}_{s,2}(\mathcal{V}^s) \in \mathbb{R}^{N \times 1}, \tag{21}$$

$$\hat{Y}^s = \hat{\mathcal{X}}^s + \tilde{\mathcal{V}}^s, \ \ \hat{Y} = \sum_{s=1}^{S} \hat{Y}^s, \tag{22}$$

where $X^s \in \mathbb{R}^{(N \times T^s \times d)}$ represents the initial node embedding in Equation 5 that does not consider the lead-lag relationship and multi-scale interaction effects. However, $\mathcal{V}^s \in \mathbb{R}^{(N \times T \times d)}$ is the spatio-temporal hypernode embedding in Equation 20 that incorporates lead-lag correlation and multi-scale information. Finally, we combine the prediction results at different scales to obtain the final prediction result.

### 3.7 OPTIMIZATION OBJECTIVES

Following StockMixer (Fan & Shen, 2024), we use the return ratio of a stock as the groundtruth. We use a combination of a pointwise regression and pairwise ranking-aware loss to minimize the MSE between the predicted and actual 1-day return ratios, while maintaining the relative order of top-ranked stocks with higher expected return for investment as follows.

$$\mathcal{L}_{MSE} = ||\hat{Y} - Y||_2^2, \ \mathcal{L}_{Rank} = \sum_{i=1}^{d} \sum_{j=1}^{d} \max(0, -(\hat{r}_i^t - \hat{r}_j^t)(r_i^t - r_j^t)), \tag{23}$$

$$\mathcal{L} = \mathcal{L}_{MSE} + \alpha \mathcal{L}_{Rank}, \tag{24}$$

where $\hat{r}^t$ and $r^t$ are the predicted and actual ranking scores, respectively, and $\alpha$ is a weight parameter.

## 4 EXPERIMENTS

### 4.1 EXPERIMENTAL SETUP

**Datasets.** We evaluate our approach using three real-world datasets from the US stock market. These datasets contain relatively complete stock-industry relationships or Wiki-based company relations. The NASDAQ and NYSE datasets (Feng et al., 2019b) include EOD data from 01/02/2013 to 12/08/2017, filtered from the respective markets. Abnormal patterns and penny stocks were removed, while preserving their representative properties, with NASDAQ being more volatile and NYSE more stable. The S&P500 dataset (Huynh et al., 2023) gathers historical price data and industry information for the companies listed in the S&P500 index from the Yahoo Finance database, covering the period from 01/04/2016 to 05/25/2022. The datasets statistics are provided in Table 4.

**Baselines.** We conduct a comparative analysis between our proposed Hermes and a range of baseline methods, which include prominent approaches in both machine learning and deep learning methods for STSF. Given that existing research has demonstrated the limited effectiveness of traditional statistical methods (Fan & Shen, 2024; Li et al., 2024), we exclude them from our comparative evaluation to focus on more competitive approaches.

### 4.2 OVERALL PERFORMANCE

We present a detailed comparison of our approach (Hermes) with four types of baseline methods in Table 1. We have the following observations: 1) For univariate methods, whether it is LSTM or

Table 1: Comparison results on public stock market datasets (measured by t-test with p-value < 0.01). The methods for comparison are mainly divided into four types: RNN (Recurrent Neural Network), GNN (Graph Neural Network), HGNN (HyperGraph Neural Network), and MLP (Multi-Layer Perceptron). Bold indicates the best (SOTA) results.

| | Model | NASDAQ | | | | NYSE | | | | S&P500 | | | |
|---|---|---|---|---|---|---|---|---|---|---|---|---|---|
| | | IC | RIC | prec@N | SR | IC | RIC | prec@N | SR | IC | RIC | prec@N | SR |
| RNN | LSTM | 0.032 | 0.354 | 0.514 | 0.892 | 0.024 | 0.256 | 0.512 | 0.857 | 0.031 | 0.186 | 0.531 | 1.332 |
| | ALSTM | 0.035 | 0.371 | 0.522 | 0.941 | 0.023 | 0.276 | 0.519 | 0.764 | 0.029 | 0.181 | 0.532 | 1.298 |
| GNN | RGCN | 0.034 | 0.382 | 0.516 | 1.054 | 0.025 | 0.275 | 0.517 | 0.932 | 0.028 | 0.175 | 0.528 | 1.359 |
| | GAT | 0.035 | 0.377 | 0.530 | 1.233 | 0.025 | 0.297 | 0.521 | 1.070 | 0.034 | 0.191 | 0.541 | 1.484 |
| | RSR-I | 0.038 | 0.398 | 0.531 | 1.238 | 0.026 | 0.284 | 0.519 | 0.098 | 0.033 | 0.200 | 0.542 | 1.437 |
| MLP | Linear | 0.019 | 0.188 | 0.505 | 0.517 | 0.015 | 0.163 | 0.497 | 0.625 | 0.016 | 0.156 | 0.520 | 0.674 |
| | StockMixer | 0.043 | 0.501 | **0.545** | 1.465 | 0.029 | 0.351 | 0.539 | 1.454 | 0.041 | 0.262 | 0.551 | 1.586 |
| HGNN | STHAN-SR | 0.039 | 0.451 | 0.543 | 1.416 | 0.029 | 0.344 | 0.542 | 1.228 | 0.037 | 0.227 | 0.549 | 1.533 |
| | ESTIMATE | 0.037 | 0.444 | 0.539 | 1.307 | 0.030 | 0.327 | 0.536 | 1.115 | 0.035 | 0.241 | **0.553** | 1.547 |
| | Hermes (Ours) | **0.044** | **0.538** | 0.535 | **2.161** | **0.032** | **0.466** | 0.549 | **1.655** | **0.050** | **0.334** | 0.539 | **2.247** |

ALSTM, their performance is inferior to that of other hybrid architectures. This strongly highlights the necessity and importance of capturing the complex industry and stock relationships in financial markets. 2) For GNNs algorithms such as RGCN, GAT, and RSR-I, their performance shows a significant improvement over RNNs, indicating that GNNs have strong stock relationship modeling capabilities. However, compared to HGNNs algorithms like STHAN-SR and ESTIMATE, GNNs perform slightly worse. This suggests that in the context of the financial market industry, using hypergraph models to unify the relationships of stocks within the industry plays a crucial role in enhancing forecasting performance. 3) The simple linear model fails due to insufficient inductive bias. While StockMixer strikes a balance between the simplicity of MLP models and the excellent performance of hybrid networks, it achieves the second-best results across most metrics. However, because MLP-based models inherently have smaller model capacities, the issue of insufficient inductive bias in StockMixer becomes more apparent when dealing with larger datasets. 4) Compared to other hypergraph algorithms that consider stock relationships within industries (such as STHAN-SR and ESTIMATE), the Hermes algorithm we propose shows significant performance improvement. This clearly demonstrates that, based on the consideration of stock relationships within industries, incorporating inter-industry lead-lag relationships and multi-scale relationships plays a crucial role in enhancing forecasting capability.

## 4.3 ABLATION STUDIES

Table 2 illustrates the unique impact of each module and contrasts them with two hypergraph-based baseline models, STHAN-SR and ESTIMATE. We have the following observations: 1) When the **multi-scale fusion module** in Section 3.5 and the **total multi-scale module** (Remove the Total Multi-Scale module, including Decomposition in Section 3.2 and Fusion in Section 3.5, respectively) are removed, the performance of the model decreases in both cases. The performance drop is more severe when the entire module is removed, indicating that while decomposition helps, the internal fusion of multi-scale information is key to further enhancing performance. 2) Ablating the lead-lag module in Section 3.4, which limits the model to only intra-industry correlations, causes a notable performance decline. This highlights the importance of modeling inter-industry lead-lag relationships. Even without this module, Hermes still outperforms STHAN-SR and ESTIMATE which only consider internal industry correlations, demonstrating the superiority of Hermes's core architecture over other hypergraph-based approaches. 3) Removing the skip-connection module in Section 3.6 also reduces performance, confirming its crucial role in facilitating information transmission and gradient flow, thereby improving the model's feature extraction capabilities.

## 4.4 EFFICIENCY ANALYSIS

In order to ensure good applicability of the model in practical applications, the runtime and memory usage of the model are crucial. We conducted an efficiency comparison analysis of Hermes and two other hypergraph-based methods, STHAN-SR and ESTIMATE—see Table 3.

Table 2: Ablation study results on public datasets NASDAQ, NYSE, and S&P500.

| Ablation | NASDAQ | | | | NYSE | | | | S&P500 | | | |
|---|---|---|---|---|---|---|---|---|---|---|---|---|
| Model Component | IC | ICIR | Prec@N | SR | IC | ICIR | Prec@N | SR | IC | ICIR | Prec@N | SR |
| w/o Multi-Scale Fusion | 0.038 | 0.464 | 0.521 | 1.374 | 0.028 | 0.421 | 0.545 | 1.354 | 0.031 | 0.248 | 0.531 | 1.517 |
| w/o Total Multi-Scale | 0.030 | 0.422 | 0.506 | 0.975 | 0.023 | 0.347 | 0.527 | 1.199 | 0.032 | 0.213 | 0.503 | 1.075 |
| w/o Lead-Lag | 0.039 | 0.472 | 0.526 | 1.451 | 0.028 | 0.420 | 0.540 | 0.225 | 0.037 | 0.201 | 0.519 | 1.628 |
| w/o Skip-Connection | 0.041 | 0.441 | 0.536 | 1.655 | 0.029 | 0.345 | 0.539 | 1.449 | 0.043 | 0.252 | 0.522 | 1.653 |
| STHAN-SR | 0.039 | 0.451 | **0.543** | 1.416 | 0.029 | 0.344 | 0.542 | 1.228 | 0.037 | 0.227 | 0.549 | 1.533 |
| ESTIMATE | 0.037 | 0.444 | 0.539 | 1.307 | 0.030 | 0.327 | 0.536 | 1.115 | 0.035 | 0.241 | **0.553** | 1.547 |
| Hermes (Ours) | **0.044** | **0.538** | 0.535 | **2.161** | **0.032** | **0.466** | 0.549 | **1.655** | **0.050** | **0.334** | 0.539 | **2.247** |

Table 3: Efficiency analysis on public stock datasets, concerning the training time, inference time (measured in milliseconds per batch under the condition of a batch size being 1), and memory usage (quantified in GB) of different models.

| Dataset | NASDAQ | | | NYSE | | | S&P500 | | |
|---|---|---|---|---|---|---|---|---|---|
| Property | STHAN-SR | ESTIMATE | Hermes | STHAN-SR | ESTIMATE | Hermes | STHAN-SR | ESTIMATE | Hermes |
| Training Time | 257 ms | 1,267 ms | 215 ms | 384 ms | 1,462 ms | 282 ms | 297 ms | 1,317 ms | 211 ms |
| Inference Time | 52 ms | 163 ms | 42 ms | 73 ms | 241 ms | 61 ms | 32 ms | 109 ms | 25 ms |
| Memory Usage | 2.15 GB | 9.62 GB | 0.98 GB | 2.82 GB | 10.25 GB | 1.14 GB | 1.23 GB | 4.52 GB | 0.54 GB |

**Runtime Analysis:** In practical application scenarios, fast response is one of the key factors. For example, in the field of financial market forecasting, timely obtaining forecasting results can provide valuable decision-making basis for investors. If the model takes too long to operate, even if the prediction accuracy is high, it will be difficult to meet the actual needs. Therefore, we evaluate the computational speed by recording the time each model takes to train and infer a sample (that is, the batch size is uniformly set to 1) with the same dataset. The results show that the Hermes model performs excellently. Compared with the other two hypergraph-based methods, its computational time is reduced. This is mainly attributed to the fact that some modules of the Hermes model adopt an efficient MLP structure in the algorithm design process. This structure can significantly improve the computational efficiency while ensuring the accuracy of forecasting results.

**Memory Analysis:** Memory usage is also an important factor to measure the efficiency of the model. Excessive memory usage will not only limit the application range of the model in different hardware environments but also may cause the system to run slowly or even crash. In this experiment, we monitored the memory usage of each model in detail during operation. The results show that the Hermes has a lower memory usage rate. This advantage primarily stems from its unique architectural design, which enables superior performance without requiring large hidden dimensions, as further validated in Section B.5 through sensitivity analysis of the Latent Space Dimension parameter. Consequently, Hermes effectively reduces memory usage while maintaining high performance, enhancing both the practicality and deployment flexibility of the model.

## 5 CONCLUSIONS

Stock time series forecasting plays a pivotal role in the modern economy, serving as a key tool for investors, financial institutions, and policymakers to make accurate decisions. However, existing forecasting methods often fall short in fully considering the lead-lag between industries and multi-scale information, resulting in limited predictive performance. To address these challenges, this study introduces the Hermes framework. By ingeniously fusing multi-scale analysis with lead-lag modules within a hypergraph network, the Hermes framework significantly boosts the performance of stock forecasting. Specifically, the hyperedge-based moving aggregation module incorporates a sliding window and utilizes dynamic temporal aggregation operations to consider lead-lag dependencies among industries. Meanwhile, the hyperedge-based multi-scale fusion module decomposes raw data at multiple scales, and then employs cross-scale, edge-to-edge message passing to integrate information from different scales while preserving the consistency of each scale. Experimental results on multiple real-world stock datasets show that Hermes outperforms existing state-of-the-art methods in both efficiency and accuracy. All datasets and code are available at `https://anonymous.4open.science/r/Hermes-E150`.

ETHICS STATEMENT

Our work exclusively uses publicly available benchmark datasets that contain no personally identifiable information. No human subjects are involved in this research.

REPRODUCIBILITY STATEMENT

The promise that all experimental results can be reproduced. We have released our model code in an anonymous repository: `https://anonymous.4open.science/r/Hermes-E150`.

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

# A  APPENDIX

## A.1  THE USE OF LARGE LANGUAGE MODELS (LLMS)

We do not use Large Language Models in our methodology and writing.

## A.2  EXPERIMENTAL SETUP

### A.2.1  DATASETS

To validate the performance of Hermes, we use three publicly available stock market datasets, with the dataset statistics provided in Table 4. These datasets provide a diverse set of stock time series, allowing us to comprehensively assess the model's effectiveness across different financial scenarios. We evaluate our approach using three real-world datasets from the US stock market. These datasets contain relatively complete stock-industry relationships or Wiki-based company relations. The NASDAQ and NYSE datasets (Feng et al., 2019b) include EOD data from 01/02/2013 to 12/08/2017, filtered from the respective markets. Abnormal patterns and penny stocks were removed, while preserving their representative properties, with NASDAQ being more volatile and NYSE more stable. The S&P500 dataset (Huynh et al., 2023) gathers historical price data and industry information for the companies listed in the S&P500 index from the Yahoo Finance database, covering the period from 01/04/2016 to 05/25/2022.

Table 4: Statistics of datasets.

| Dataset Property | NASDAQ | NYSE | S&P500 |
|---|---|---|---|
| # Stocks | 1026 | 1737 | 474 |
| # Industries | 113 | 130 | 11 |
| # Technical Indicators | 5 | 5 | 5 |
| Start Time | 13-01-02 | 13-01-02 | 16-01-04 |
| End Time | 17-12-08 | 17-12-08 | 22-05-25 |
| Total Time Steps | 1281 | 1281 | 1611 |
| Train Time Steps | 756 | 756 | 1006 |
| Validate Time Steps | 252 | 252 | 253 |
| Test Time Steps | 273 | 273 | 352 |

### A.2.2  IMPLEMENTATION DETAILS

For fair comparison, all samples are generated by moving a 16 time steps lookback window. All experiments with Hermes are conducted using PyTorch (Paszke et al., 2019) in Python 3.9, and are executed on a server featuring an Intel(R) Xeon(R) Platinum 8358 CPU and an NVIDIA Tesla-A800 GPU. We use ADAM (Kingma & Ba, 2015) with an initial learning rate of 5e - 3. The "*Drop Last*" issue is reported by several researchers (Qiu et al., 2024; 2025). That is, in some previous works evaluating the model on test set with drop-last=True setting may cause additional errors related to test batch size. In our experiment, to ensure fair comparison in the future, we set the drop last to False for all baselines to avoid this issue.

### A.2.3  METRICS

Previous studies use different evaluation metrics, making it challenging to perform a comprehensive comparison of various methods. To provide a thorough assessment of the methods performance, we use four of the most commonly applied and reliable metrics: **IC** and **ICIR**, which are rank-based evaluation metrics; **Prec@N**, which measures accuracy; and **SR**, which is return-based.

- **Information Coefficient (IC):** IC is a coefficient that shows how close the prediction is to the actual result, computed by the average pearson correlation coefficient.

$$IC_t = \frac{1}{N} \frac{(\hat{Y}_t - mean(\hat{Y}_t))^T (Y_t - mean(Y_t))}{std(\hat{Y}_t) \cdot std(Y_t)}, \tag{25}$$

where $Y_t$ are the raw stock price trends and $\hat{Y}_t$ are the model predictions at each timestamp. We report the average IC over all test dataset.

- **Information Ratio-based Information Coefficient (ICIR):** The information ratio of the IC metric, calculated by:

$$ICIR = \frac{mean(IC)}{std(IC)}. \tag{26}$$

- **Precision@N (Prec@N):** Prec@N evaluates the precision of the top N short-term profit predictions from the model. This way, we assess the capability of the techniques to support investment decisions. For example, when N is 10, and the labels of 6 among these top 10 predictions are positive, then the Prec@10 is 60%.

$$Prec@N = \frac{TP@N}{N}, \tag{27}$$

where $TP$ is the number of true positive classifications. In this study, $N$ is set to 10.

- **Sharpe Ratio (SR):** Sharpe ratio (Sharpe, 1994) measures the profitability of the investment method and take into account risk.

$$SR = \frac{E_t[\rho_t - \rho_F]}{\sqrt{\mathrm{var}_t(\rho_t - \rho_F)}}, \tag{28}$$

where $\rho_t = \frac{p_t}{p_{t-1}} - 1$ is the return on the portfolio and $p_F$ is the return on the risk-free asset which is always 0.

### A.2.4 BASELINES

We conduct a comparative analysis between our proposed Hermes and a range of baseline methods, which include prominent approaches in both machine learning and deep learning methods for STSF. Given that existing research has demonstrated the limited effectiveness of traditional statistical methods (Fan & Shen, 2024; Li et al., 2024), we exclude them from our comparative evaluation to focus on more competitive approaches.

- **LSTM** (Hochreiter & Schmidhuber, 1997) refers to the standard LSTM model, which operates on sequential data, including closing prices and moving averages over 5, 10, 20, and 30 days, to generate a sequential embedding. A fully connected (FC) layer is then employed to predict the return ratio.

- **ALSTM** (Feng et al., 2019a) integrates adversarial training and stochastic simulation into an enhanced LSTM model, allowing it to more effectively capture market dynamics.

- **RGCN** (Li et al., 2021) introduces a model that captures both positive and negative correlations among stocks using a correlation matrix computed from historical market data. The LSTM mechanism in RGCN layers helps mitigate over-smoothing issues when predicting overnight stock price movements.

- **GAT** (Veličković et al., 2017) employs Graph Attention Networks (GAT) to aggregate stock embeddings, which are encoded by a GRU, on the stock graph. By combining the advantages of Graph Neural Networks (GNN) and attention mechanisms, it enhances performance in large-scale graph settings.

- **RSR** (Feng et al., 2019b) introduces a novel neural network-based framework called Relational Stock Ranking, which addresses the stock forecasting problem in a learning-to-rank approach. Additionally, it proposes a new component in neural network modeling, called Temporal Graph Convolution, which is designed to explicitly capture the domain knowledge of stock relationships in a time-sensitive manner.

- **STHAN-SR** (Sawhney et al., 2021) introduces a novel Spatio-Temporal Hypergraph Attention Network that models inter-stock relationships of different types and strengths as a hypergraph for stock ranking. It combines temporal Hawkes attention with spatial hypergraph convolutions through hypergraph attention to capture both the correlations in the movements of related stocks and the temporal evolution of their historical features.
- **ESTIMATE** (Huynh et al., 2023) integrates temporal generative filters within a memory-based shared parameter LSTM network, enhancing the model's ability to learn temporal patterns for each stock. Furthermore, it introduces attention hypergraph convolutional layers that utilize the wavelet basis—a convolution approach that leverages the polynomial wavelet basis to streamline message passing and focus on localized convolutions.
- **Linear** (Zeng et al., 2023) employs a straightforward approach by utilizing only fully connected layers to predict the final price. This method does not incorporate any advanced architectures or temporal dependencies, relying solely on basic linear transformations to generate the forecasting. Despite its simplicity, it serves as a baseline model for comparison with more complex models.
- **StockMixer** (Fan & Shen, 2024) introduces a lightweight yet effective MLP-based architecture for stock price forecasting. The model incorporates three key components—indicator mixing, time mixing, and stock mixing—which work together to capture the intricate correlations within the stock data.

## B  RELATED WORK

### B.1  TRADITIONAL STOCK TIME SERIES FORECASTING

Early stock forecasting methods rely mainly on statistical learning techniques to capture patterns and relationships in time series (Ye et al., 2016; Ariyo et al., 2014; Gupta & Dhingra, 2012; Elton et al., 2009). One of the most widely used techniques is the Autoregressive Integrated Moving Average (ARIMA) (Ariyo et al., 2014) model, which stabilizes data by combining Autoregressive components, Moving Average, and differencing, thereby modeling time series. Another commonly used traditional method is the Hidden Markov Model (HMM) (Gupta & Dhingra, 2012), which is used to describe unobservable states in the market and can detect changes in market conditions, making it suitable for analyzing market transitions. Although these methods provide predictive capabilities, they exhibit significant limitations when dealing with nonlinear, dynamic changes, and sudden events in financial markets. As a market environment becomes more complex, traditional statistical methods increasingly fail to provide sufficient predictive accuracy and robustness.

With the rapid development in machine learning, machine learning methods for stock time series forecasting have emerged (Alkhatib et al., 2013; Nugroho et al., 2014; Kim, 2003). For example, Support Vector Machines (SVMs) (Kim, 2003) map data to high-dimensional spaces using kernel functions and find optimal separating hyperplanes in those spaces, which allows them to capture complex relationships and dynamics in financial markets. However, SVMs are highly sensitive to parameter selection and often struggles to capture long-term dependencies. Overall, although machine learning methods show considerable potential in stock forecasting, they still rely on manual feature engineering and model design. Moreover, these traditional stock forecasting methods have not delved into the exploration of lead-lag relationships and multi-scale modeling.

### B.2  DEEP LEARNING STOCK TIME SERIES FORECASTING

With the rise of deep learning, notable progress has occurred in stock forecasting. This is mainly because deep neural networks can model complex nonlinear patterns. This is mainly because deep neural networks can model complex nonlinear patterns. Some studies use recurrent neural networks (RNNs) (Nelson et al., 2017; Qin et al., 2017; Akita et al., 2016; Rahman et al., 2019) to capture the historical development in individual stock prices and predict their short-term trends. For example, Long Short-Term Memory (LSTM) networks combined with historical stock data and text information are used to predict stock prices (Akita et al., 2016). In addition, due to the significant progress of Multi-Layer Perceptrons (MLPs) architecture in the general time series field, the DLinear (Zeng et al., 2023), CycleNet (Lin et al., 2025), and SparseTSF (Lin et al., 2024) show the benefits of this simple architecture for time series forecasting. Increasingly may proposals target stock forecasting performance by improving the MLPs architecture. For example, a series of studies (Fan & Shen,

2024; Lazcano et al., 2024; Tashakkori et al., 2024) enhance the learning ability of simple MLPs structures by utilizing the MLP-Mixer backbone, thereby improving forecasting accuracy. Next, the Transformer model has become increasingly popular due to the ability of its self-attention mechanism to capture long-term dependencies. For example, Master (Li et al., 2024) proposes a novel stock transformer for stock price forecasting to effectively capture stock correlation.

Some deep learning models have begun to explore lead-lag relationships and multi-scale modeling. For instance, StockMixer (Fan & Shen, 2024) engages in multi-scale modeling by segmenting the original time series into subsequence-level patches and mixing features at different scales. However, this approach lacks effective integration of information across scales; it merely concatenates information from different scales without thorough fusion. On the other hand, ESTIMATE (Huynh et al., 2023) takes into account lead-lag relationships through data-driven detection of leading and lagging time series clusters. Nonetheless, this method falls short in capturing fine-grained lead-lag dynamics, as clustering can only reveal global lead-lag relationships. Distinctive from the aforementioned approaches, Hermes employs a sliding window technique and utilizes dynamic temporal aggregation operations to consider fine-grained lead-lag dependencies among stock industries. Moreover, it features a multi-scale fusion module, which utilizes a cross-scale, egde-to-edge message-passing to integrate information from different scales while maintaining the consistency of each scale.

While these deep learning methods have advanced stock forecasting, the complexity of stock data and the need to account for intricate relationships among stocks have led researchers to explore new methods based on hypergraphs.

### B.3 HYPERGRAPH-BASED STOCK TIME SERIES FORECASTING

Traditional graphs have limitations when it comes to modeling higher-order multivariate relationships, as they can only represent relations between pairs of nodes. To address this issue, Hypergraph Neural Networks (HGNNs) (Feng et al., 2019c) were introduced. Hypergraphs extend the concept of simple graphs to enable the capture of relationships among multiple stocks (Chen et al., 2020). Hypergraphs have gained increasing attention and application across various fields. Hyperedge in a hypergraph represents a set of vertices, making them suitable for modeling non-pairwise relationships (Luo et al., 2014; Sawhney et al., 2020). For example, HGAM (Li et al., 2022) is a hypergraph-based reinforcement learning method for stock portfolio selection. THGNN (Xiang et al., 2022) is a temporal and heterogeneous graph neural network model that aims to learn dynamic relationships by using two-stage attention mechanisms. STHAN-SR (Sawhney et al., 2021) enhances corporate relevance based on Wiki data and uses hypergraph convolution to propagate information from higher-order neighbors. DHSTN (Liao et al., 2024) is a dynamic hypergraph network for learning spatio-temporal relations of stocks. Dynamic hypergraphs are generated adaptively using (Graph Attention Networks) GATs to capture time-varying higher-order spatial relationships among stocks. The latest method, ESTIMATE (Huynh et al., 2023), employs hypergraphs to capture non-pairwise correlations, utilizing temporal generative filters to identify individual patterns for each stock.

Unlike the above methods, we propose a new hypergraph-based stock forecasting model that specifically takes into account the multi-scale nature of stock data, enabling it to capture the complexity of market dynamics more comprehensively. Furthermore, our approach places particular emphasis on the lead-lag relationships between financial industries, where the changes in one industry may precede or lag behind those in another. By capturing these key relationships more fully, we aim to improve forecasting accuracy.

### B.4 ILLUSTRATION FIGURES

### B.5 HYPERPARAMETER SENSITIVITY

This experiment addresses question RQ3 on the hyperparameter sensitivity. Due to space limitations, we focus on the most important hyperparameters—see Figure 5.

**Lookback window length $T$:** We examine the predictive performance of Hermes as we adjust the lookback window length, $T$, illustrated in Figure 5a. Consistently across all datasets, a balanced window length emerges as the most effective. Window lengths that are too short result in a swift decline in performance due to inadequate information, whereas excessively lengthy sequences also

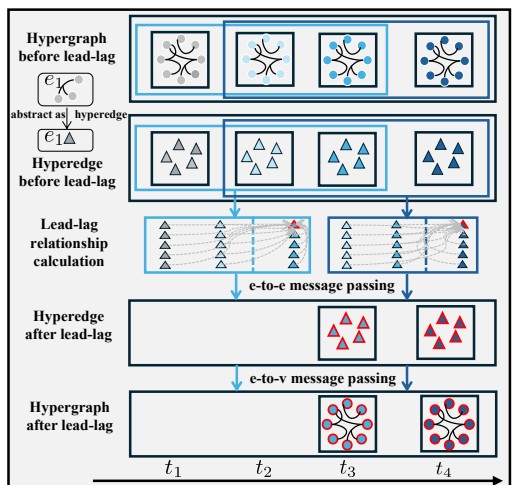

Figure 3: The specific lead-lag interaction process within the window, with a lead-lag step of 3 as an example.

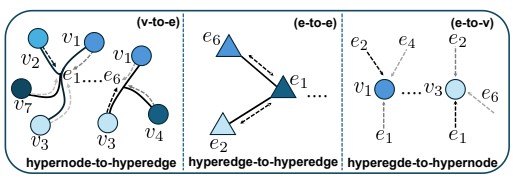

Figure 4: Three types of message passing: hypernodes for stocks and hyperedges for industries.

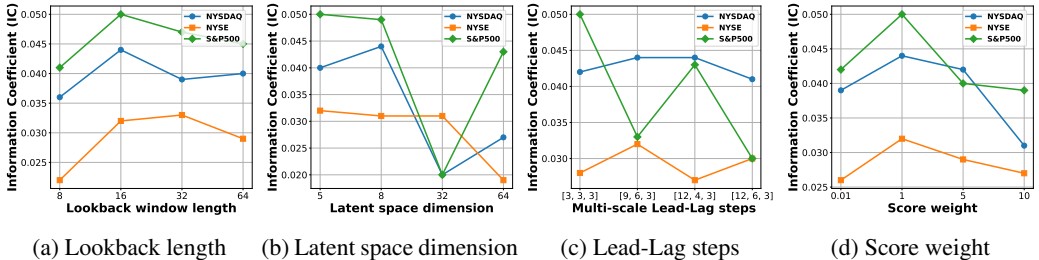

| (a) Lookback length | (b) Latent space dimension | (c) Lead-Lag steps | (d) Score weight |

Figure 5: Parameter sensitivity studies of main hyper-parameters in Hermes.

underperform because of the absence of timely information benefits and the heightened learning expenses for stocks.

**Latent space dimension** $d$: In Section 3.2, we enhance the representational capacity of indicators by mapping them into a high-dimensional hidden space $d$, thereby facilitating the extraction of more intricate hidden relationships. In Figure 5b, we explore the impact of different hidden dimensions $d$ and observe that datasets exhibit optimal performance at varying $m$ values. Notably, when $d$ is set to smaller values such as 5 or 8, models tend to achieve good performance; however, as $d$ increases, models become prone to overfitting, leading to a decline in performance. Furthermore, selecting smaller $d$ not only improves model performance but also effectively reduces the number of model parameters.

**Multi-scale Lead-Lag steps** $w^1, w^2, \cdots, w^S$: In Section 3.4, for time series at each scale, we study the lead-lag correlations within $w^s$ time steps to better extract complex dependencies in the time series, thereby improving forecasting performance. Here, we select three scales and specify the lead-lag time steps as $w^1$, $w^2$, and $w^3$, respectively. As shown in Figure 5c, we find that different datasets have variations in their multi-scale lead-lag time steps when achieving optimal performance. This is because different datasets possess unique characteristics, and their cyclicity (fine-grained scale) and trend (coarse-grained scale) may change due to external factors. Therefore, we recommend adjusting this parameter specifically when dealing with new datasets.

**Score weight** $\alpha$: Besides, we adopt the score weight in section 3.7 to trade off the MSE loss and the ranking-aware loss—see Figure 5d. Experimental results show that all datasets achieve optimal performance and exhibit high stability when the score weight is set to 1. Therefore, we recommend setting this parameter to 1 in future research to obtain more optimized results.

