# OpenReview forum: "Multi-Scale Spatial-Temporal Hypergraph Network with Lead-Lag Structures for Stock Time Series Forecasting"
_ICLR.cc/2026/Conference — ICLR 2026 Conference Withdrawn Submission_

### Official Review · Reviewer_H5NL · 2025-10-26

**Soundness:** 3
**Presentation:** 3
**Contribution:** 3
**Rating:** 4
**Confidence:** 3

**Summary:**

The paper introduces Hermes, a multi-scale spatial-temporal hypergraph network for stock time series forecasting. The model aims to jointly model inter-industry lead-lag structures and multi-scale temporal dependencies. It incorporates a hyperedge-based moving aggregation module and a multi-scale fusion mechanism to capture both spatial and temporal dynamics. Experiments on NASDAQ, NYSE, and S&P500 datasets demonstrate significant improvements over several baselines (RNN-, GNN-, HGNN-, and MLP-based models).

**Strengths:**

1.This work introduces a unified framework that jointly models lead-lag relations and multi-scale temporal patterns using hypergraph structures, which is an original and well-motivated contribution.

2.Hermes shows strong empirical performance, achieving consistent improvements on several key financial forecasting metrics (IC, ICIR, Prec@N, and SR) across multiple benchmark datasets.

3.The paper is clearly written, with well-structured methodology and rigorous mathematical reasoning that supports the soundness of the proposed framework.

**Weaknesses:**

1.The paper mainly compares against classical RNN, GNN, HGNN, and MLP models, but omits transformer-based time-series forecasting methods, such as PatchTST[1] and TimeMixer[2]. Given that Hermes also models multi-scale temporal dependencies, these methods are highly relevant and should be included for an up-to-date comparison.

[1] A Time Series is Worth 64 Words: Long-term Forecasting with Transformers (ICLR 2023)
[2] TimeMixer: Decomposable Multiscale Mixing for Time Series Forecasting (ICLR 2024)



2.All experiments are conducted on U.S. stock market data. The absence of cross-market or temporal robustness tests makes it unclear whether Hermes generalizes to other markets or economic regimes.

3.The model uses a fixed industry matrix H, where each stock belongs to a predefined industry, and this structure does not change over time. While the sliding-window module captures how stock features evolve within an industry, the relationships between industries remain static. As a result, Hermes only models temporal dynamics under a static relational structure and fails to reflect shifting dependencies between industries in different market conditions. This limitation may reduce its reliability and adaptability when applied to real-world financial markets, where inter-industry relationships are highly dynamic.

4.Although the paper models lead-lag structures, it lacks economic or causal interpretation of these relationships. It remains unclear whether the learned dependencies reflect real inter-industry dynamics or mere statistical correlations. Some visualization or Granger-causality-style validation would strengthen the credibility of this claim.

**Questions:**

1.Why were recent state-of-the-art time-series forecasting models (e.g., PatchTST, TimeMixer) not included in the experimental comparison?

2.Can Hermes generalize to other regions or market periods (e.g., non-U.S. data or post-COVID regimes)?

3.Have the authors considered employing a time-varying or learned relational structure (e.g., dynamic graphs based on co-movement or correlation) rather than a fixed industry matrix?

---

### Official Review · Reviewer_RX61 · 2025-10-29

**Soundness:** 3
**Presentation:** 3
**Contribution:** 3
**Rating:** 6
**Confidence:** 4

**Summary:**

This paper addresses the limitations of existing hypergraph-based models for Stock Time Series Forecasting (STSF), which tend to overlook two critical aspects of financial markets: **inter-industry lead-lag interactions** and **multi-scale information**. The authors propose Hermes, a novel spatial-temporal hypergraph network, designed to capture these correlations more comprehensively. The core of the framework integrates two primary modules into the hypergraph structure: **(1) Hyperedge-based Moving Aggregation**: This module utilizes a sliding window and dynamic temporal aggregation to model the non-simultaneous, causal (lead-lag) relationships that exist between different financial industries. **(2) Hyperedge-based Multi-scale Fusion**: To capture patterns at different time granularities (e.g., short-term cycles versus long-term trends) , the input time series is first decomposed into multiple scales. The fusion module then integrates this multi-scale information using a novel cross-scale, edge-to-edge message passing mechanism, which preserves the consistency of information at each scale while facilitating interaction between them. The architecture successfully unifies these spatio-temporal dynamics and demonstrates superior performance, in both accuracy and efficiency, across multiple real-world stock datasets.

**Strengths:**

- The framework is commendably designed to solve two core, long-standing challenges in financial time series: causality/lead-lag and multi-scale analysis.
- The hyperedge-based moving aggregation module is an elegant solution for modeling dynamic industrial relationships. By applying the sliding window mechanism to hyperedges (industries), the model captures industry-to-industry dependencies, which are arguably more stable and meaningful in financial markets than stock-to-stock interactions alone.
- The design to first decompose the data into different scales and then employ a cross-scale, edge-to-edge message passing mechanism  is a robust way to merge diverse information.
- The overall presentation is neat and easy to follow.

**Weaknesses:**

- The proposed approach captures temporal correlations within sliding windows but does not establish causal directionality. No comparison with Granger causality, neural causal inference, or counterfactual testing is provided.
- Key components are described using pseudo-code-like equations rather than rigorous mathematical formulations. Theoretical justification for the lead-lag and multi-scale modules is lacking.
- The paper does not compare against non-hypergraph methods specifically designed for temporal causality or lead-lag modeling, making it difficult to assess whether the hypergraph structure is essential to the observed gains.
- The hypergraph construction is fundamentally constrained by a global-shared stock-industry relationship matrix. Financial relationships are often dynamic and cross traditional industrial boundaries. Relying on a static classification to define the initial hyperedges may limit the model's ability to discover truly emergent and transient, data-driven high-order correlations. *I present this as a discussion point rather than a criticism.*

**Questions:**

Please see weakness above.

---

### Official Review · Reviewer_DXf1 · 2025-10-31

**Soundness:** 3
**Presentation:** 3
**Contribution:** 2
**Rating:** 2
**Confidence:** 3

**Summary:**

This paper proposes Hermes, a multi-scale spatial-temporal hypergraph network that aims to capture inter-industry lead-lag correlations and multi-scale dependencies for stock time series forecasting. The overall presentation is clear and the paper is technically sound. However, the topic of stock prediction has been extensively studied over decades, and the contribution here seems rather incremental—mainly achieving better results on well-worn benchmark datasets without offering truly novel insights into financial modeling. Moreover, several core conceptual points remain insufficiently discussed.

**Strengths:**

S1. The paper is generally well-organized and easy to follow.

S2. The motivation is clearly stated, and the overall narrative flows smoothly.

S3. The proposed model is competently implemented, with comprehensive experimental comparisons.

**Weaknesses:**

**Concerns**

C1. The paper places great emphasis on correlations among industries and stocks. However, it is not clear whether these correlations are causal or merely coincidental. The problem of spurious correlations in financial time series has been extensively discussed in causal inference and econometrics literature. Without deeper validation or domain-based justification, it remains uncertain whether the discovered “industry correlations” truly reflect economic interdependence or are statistical artifacts. The authors should clarify how the model ensures that the identified relationships are robust and meaningful rather than coincidental.

C2. The motivation for Hermes relies on the assumption that meaningful lead-lag correlations exist and can be exploited for forecasting. However, the paper does not provide any concrete case study or example illustrating such lead-lag phenomena within the benchmark datasets. Presenting one or two specific examples—where a leading industry consistently predicts another—would strongly substantiate the main hypothesis and demonstrate the model’s interpretability.

C3. The notion of temporal lag is central to the model’s design, but its practical implications are not well explored. Specifically, how does the size of the prediction window affect model performance? In financial markets, larger windows often reduce predictive accuracy, yet true lead-lag effects may unfold over extended horizons. The current experiments overlook this interplay.

Moreover, the appendix (page17) briefly notes that different datasets exhibit varying lag behaviors—this actually exposes a key limitation: each stock or sector may possess distinct lag features. A fixed hyperparameter for lag size may therefore fail to generalize. The authors should analyze the relationship between lag-window size, stock responsiveness, and predictive accuracy in more detail.

**Overall Suggestion**

While the model is technically well implemented, the paper’s core contribution remains underdeveloped. To make the work more meaningful, the authors might consider reframing the study to focus not merely on forecasting accuracy, but on uncovering and characterizing lagged inter-industry dependencies—a perspective that could have more value for quantitative finance. Ablation studies and routine comparisons could be moved to the appendix, freeing space to elaborate on this central idea.

**Questions:**

Please mainly respond to three concerns.

---

### Official Review · Reviewer_z4Ft · 2025-10-31

**Soundness:** 3
**Presentation:** 2
**Contribution:** 2
**Rating:** 2
**Confidence:** 4

**Summary:**

This paper proposes a spatial-temporal hypergraph network framework Hermes with a hyperedge-based moving aggregation module and hyperedge-based multi-scale fusion module for stock time series forecasting, which can consider lead-lag dependencies and multi-scale information. Experiments demonstrate the effectiveness of Hermes.

**Strengths:**

1.Integrating moving aggregation and multi-scale fusion in spatial-temporal hypergraph network for considering both lead-lag and multi-scale industry correlations is interesting.

2.The effectiveness of Hermes is supported by extensive experiments.

**Weaknesses:**

1.The limited innovation in the design of Hermes makes it an incremental work. Its core modules are heavily inspired by existing paradigms: The hyperedge-based moving aggregation module mirrors the 'patching' paradigm from recent attention-based time series forecasting, while the hyperedge-based multi-scale fusion employs a standard adaptive graph learning method. The contribution is more of an engineering integration rather than novel contributions. The authors should emphasize their main contribution.

2.Section 3.4 is confusing and difficult to follow, making it hard to quickly grasp the main ideas. The authors claim that the lead-lag relationship matrix is based on the relationship between the first few points and the last point within the window (on line 276), which conflicts with Formulas 10, 11, and 12. Even worse, the provided code (see class HyperGraph_LeadLag) implements a third, different logic. This discrepancy makes the module unclear and irreproducible. The authors should further clarify the design of the hyperedge-based moving aggregation module to avoid misunderstanding.

3.There are several errors in the notations and formulas. For example, on line 191, the formula for 1-day return ratio is wrong. On Formula 23, the upper bound of the summation is wrong. The authors should review the paper thoroughly.

4.The paper has some weaknesses in the experiments, which are not convincing enough:

(1)Considering Hermes is a hypergraph-based model, more hypergraph-based models in general time series forecasting should be compared to further validate the effectiveness of Hermes, e.g., Ada-MSHyper [1].

(2)In Section 4.3, the authors claim that the internal fusion of multi-scale information is key to enhancing performance, as “w/o Total Multi-Scale” performs worse than “w/o Multi-Scale Fusion” (on line 419). The claim is based on flawed reasoning, as the comparison fails to isolate the fusion module. The authors claim that Hermes outperforms the other two hypergraph-based baseline models even without the lead-lag module. However, the superiority is not obvious from Table 2 and the authors should quantify the comparison. Considering removing the skip-connection module would reduce performance, the authors should compare the “w/o Skip-Connection” against the baseline using only initial node embedding to verify whether the modeling actually improves forecasting or not.

(3)In Section 4.4, the authors should compare Hermes with MLP-based models, which is vital to evaluate the performance-efficiency trade-off of Hermes against simpler models.

[1]Shang Z, Chen L, Wu B, et al. Ada-MSHyper: Adaptive multi-scale hypergraph Transformer for time series forecasting. NIPS 2024.

**Questions:**

1.What does the L mean on line 318?

2.On Formula 19, the normalization of matrix B seems incorrect. Why not use -0.5 power of matrix D for normalization？

---

### Note · Authors · 2025-11-16

I have read and agree with the venue's withdrawal policy on behalf of myself and my co-authors.